# FDGen: A Fairness-Aware Graph Generation Model

**Zichong Wang** [1]  **Wenbin Zhang** [1] [†]

## Abstract

Graph generation models have shown significant potential across various domains. However, despite their success, these models often inherit societal biases, limiting their adoption in real-world applications. Existing research on fairness in graph generation primarily addresses structural bias, overlooking the critical issue of feature bias. To address this gap, we propose FDGen, a novel approach that defines and mitigates both feature and structural biases in graph generation models. Furthermore, we provide a theoretical analysis of how bias sources in graph data contribute to disparities in graph generation tasks. Experimental results on four real-world datasets demonstrate that FDGen outperforms state-of-the-art methods, achieving notable improvements in fairness while maintaining competitive generation performance.

## 1. Introduction

Graphs naturally appear in many real-world scenarios, from social network analysis (Grover et al., 2019), financial markets (Wang et al., 2023a), and recommendation systems (Wang et al., 2019) to the Internet of Things (Kong et al., 2023). In this context, deep learning on graph-structured data has attracted increasing attention and inspired many graph learning frameworks in recent years (Kang et al., 2022; Wang et al., 2024c; 2025h). Among them, graph generation models have become crucial components of the graph machine learning framework, serving purposes such as data augmentation (Chakrabarti & Faloutsos, 2006), anomaly detection (Akoglu et al., 2008), and enabling privacy-preserving data sharing (Kose & Shen, 2024b). Thus, creating synthetic graphs with graph generative models becomes instrumental in applications over interconnected systems.

Despite showing promising results, most graph generation models suffer from fairness issues (Wang et al., 2023c), raising ethical and societal concerns, particularly in high-stake decision-making scenarios such as healthcare (Yu et al., 2020), credit scoring (Shumovskaia et al., 2020), and crime prediction (Wang et al., 2025b). Consider a credit scoring scenario where financial institutions need synthetic data for partner collaboration. While graph generation models can protect individual privacy by creating synthetic graphs from real data, they may amplify existing biases (Wang & Zhang, 2024). For example, if the original data shows ethnic clustering in financial relationships, the generated graphs often strengthen these patterns, creating denser connections between people of the same ethnicity. These amplified structural biases then affect downstream financial decisions (Wang et al., 2025a).

To this end, preliminary efforts have been made to explore fairness within graph generation models. For instance, FairWire (Kose & Shen, 2024b) designs a fairness regularizer and leverages the proposed fair regularizer in a generative model to mitigate graph structure bias. However, FairWire, like existing fair graph generation models (Zheng et al., 2024), focuses solely on addressing structural bias, where nodes connect predominantly with neighboring nodes sharing the same sensitive attributes. For example, in the generated graph shown in Figure 1 (b), the central male node $v_1$, compared with the original graph in Figure 1(a), maintains connections with male nodes $v_2$ and $v_3$, adds a new connection with male node $v_4$, and loses its original connections with female nodes $v_5$ and $v_6$, leading to significantly higher recommendation rates within sensitive groups, potentially causing social segregation (Hofstra et al., 2017). However, all existing methods overlook feature bias, which arises when the generated non-sensitive attributes exhibit distributional disparities across subgroups. Still in Figure 1, the original graph in Figure 1 (a) depicts equal incomes across gender groups, whereas the generated graph in Figure 1 (b) shows male nodes with higher incomes than female nodes ($370,000 vs. $250,000), leading to biased downstream applications where models trained on this synthetic data learn to expect higher incomes from male applicants. In addition, feature bias is closely intertwined with structural bias, as node features propagate through the network structure. Therefore, it cannot be overlooked when aiming for

[1]Knight Foundation School of Computing and Information Sciences, Florida International University, Miami, USA. [†] Correspondence to: Wenbin Zhang <wenbin.zhang@fiu.edu>.

*Proceedings of the 42nd International Conference on Machine Learning*, Vancouver, Canada. PMLR 267, 2025. Copyright 2025 by the author(s).

truly fair graph structure generation.

It is therefore of social importance to account for multiple biases to generate fair graphs, as shown in Figure 1 (c). Specifically, $v_1$ maintains balanced connections with both male ($v_2$, $v_3$) and female neighbors ($v_5$, $v_6$), demonstrating equal connectivity across gender groups. Additionally, while preserving inherent group-specific characteristics, the node attributes, such as income, show similar distributions across different groups, eliminating feature bias. To achieve such fair graph generation, several challenges need to be addressed: **i) Difficulty in measuring feature bias:** Existing fair graph generation models primarily address structural bias by using statistical comparisons of inter-group connectivity. However, measuring node feature bias presents a critical challenge, as real-world graphs contain diverse types of node features (*e.g.*, categorical zip codes, continuous income values) with different scales and distributions that cannot be directly compared using simple statistical measures. **ii) Difficulty in mitigating multiple types of bias during generation:** Unlike independent and identically distributed (IID) data, graph generation requires ensuring fairness in both node features and graph structure. Furthermore, node features influence the formation of graph structure, and the graph structure affects the generation of node features. This interdependence makes it challenging to design universal fairness constraints that can effectively reduce both types of bias simultaneously. **iii) Difficulty in distinguishing group identity information:** While we aim to mitigate unfair differences between groups, we need to preserve legitimate group-specific characteristics. For instance, while we seek equal treatment in non-sensitive attributes like income or credit scores, we should maintain natural group differences in physical characteristics (*e.g.*, facial hair in males). The key challenge is to distinguish between biases that require mitigation and inherent group characteristics that should be maintained, thereby ensuring generated graphs that are both fair and realistic.

To tackle the aforementioned challenges, we propose a novel framework, FDGen (short for **F**air **D**iffusion for **G**raph **Gen**eration) to ensure the fairness of the graph generation model. *To the best of our knowledge, this is the first work to simultaneously mitigate multisource biases arising from sensitive attributes in graph generation models.* Specifically, we first carry out a theoretical analysis investigating the sources of bias in the graph generation model. Guided by the theoretical findings, we proposed a novel fairness regularizer, which can be interpreted as encouraging each node to weightly aggregate representations of other nodes with different sensitive attributes of the central node and weightly subtract representations of other nodes with the same sensitive attribute, which can alleviate over- association of the learned representation with sensitive attributes, resulting in fair representations with good utility ensured by

smoothness. Based on this, we design a new diffusion-based fair graph generation framework that leverages the proposed regularizer to achieve fair graph generation.

Our main contributions can be summarized as follows: **i) Problem.** We formalize the fair graph generation problem and identify unique challenges motivated by real applications. **ii) Framework.** We proposed a novel fairness-aware graph generation model to mitigate structural bias and feature bias during the graph generation while guaranteeing graph quality and enhancing fairness. **iii) Experimental Evaluation.** We conduct extensive experiments on four real-world datasets, demonstrating the superior performance of the proposed method over other state-of-the-art fairness methods.

## 2. Related Work

**Synthetic Graph Generation.** Generating synthetic graphs has been a longstanding research topic (Bojchevski et al., 2018; Kong et al., 2023), with deep generative models proving particularly successful (Li et al., 2019). Existing approaches include one-shot methods, often based on VAEs (Simonovsky & Komodakis, 2018; Liu et al., 2018) or GANs (De Cao & Kipf, 2018; Bojchevski et al., 2018), which generate all edges simultaneously from latent embeddings but may degrade structural fidelity due to independence assumptions. In contrast, autoregressive models generate graphs by sequentially adding nodes and edges. These approaches, implemented through recurrent networks (Dai et al., 2020) or reinforcement learning (You et al., 2018), capture complex structural patterns and allow constraints during generation, but remain sensitive to node ordering (Huang et al., 2022). More recently, diffusion-based methods have emerged (Liu et al., 2019; Niu et al., 2020; Chen et al., 2023; Li et al., 2023), defining a Markov chain of diffusion steps that gradually add noise to data, then learn to reverse the inference path to generate data from noise (Sohl-Dickstein et al., 2015). However, fairness remains unexplored in synthetic graph generation, limiting these models' use in high-stakes scenarios. This has created a need to develop fairer graph generation methods.

**Fairness-aware Graph Generation Model.** Fairness is a widely-existing issue of graph learning systems and has received increasing research attention in recent years (Wang et al., 2025d;c; 2024a; Wang & Zhang, 2024; Zhang et al., 2024; Zhu et al., 2024). Despite these advances, most existing methods focus on classification tasks, leaving the domain of graph generation models largely unexplored (Zhang et al., 2025). To this end, a small number of works (Rahman et al., 2019; Wang et al., 2023c; Zheng et al., 2024) have started to investigate fairness in graph generation models, which can be divided into two categories: i) Fair link prediction and ii) Fair graph structural generation. Specif-

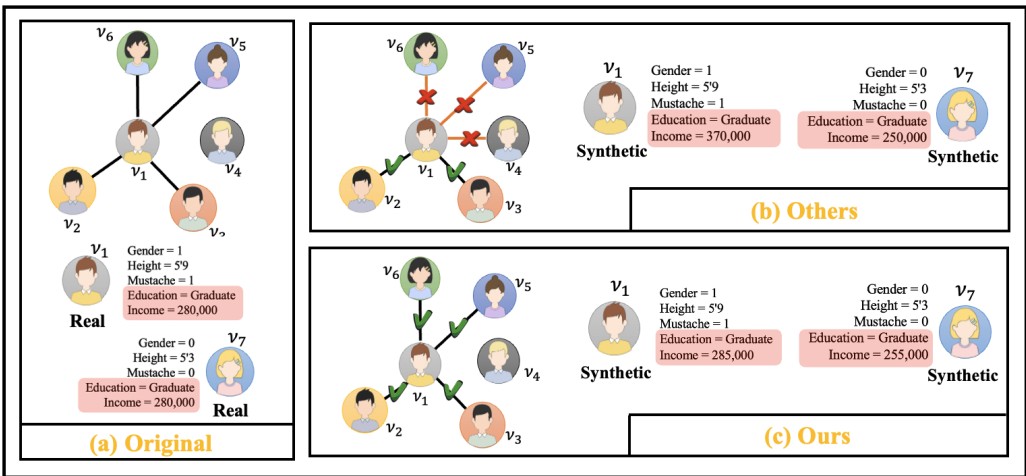

*Figure 1.* A toy example of biases in graph generation is shown, where black lines represent original edges, gray dashed lines indicate removed inter-group edges, and orange lines denote newly added intra-group edges.

ically, fair link prediction focuses on achieving unbiased predictions of dyadic relationships among graph nodes. For instance, FAIRLP (Li et al., 2022) was designed to modify the training graph to balance the distribution of intra-group and inter-group links while preserving the network characteristics of the graph. This enhancement improves the representation of underrepresented groups to achieve fair link prediction. Unlike fair link prediction, fair graph structural generation measures graph structural bias at the graph level and aims to reduce the distribution difference between the generated graph and the original graph's inter- and intra-group edges. For instance, FairGen (Zheng et al., 2024) achieves fair graph structural generation by incorporating parity constraints to minimize the difference in reconstruction loss between the generated graph and the original graph across different subgroups. However, these methods, focus solely on ensuring fairness in graph structure generation, neglecting potential biases in node information generation, which are crucial for achieving truly fair graph generation.

In contrast to existing work, this paper proposes a fair graph generation model that addresses both graph structural bias and feature bias, with its design informed by theoretical analysis. In addition, the bias mitigation approach is flexible, allowing it to be applied in both link prediction models and generative models.

## 3. Notation

In this paper, we formalize the graph generation problem in the context of an undirected graph $\mathcal{G} = (\mathcal{V}, \mathcal{E}, \mathbf{X})$, it would contain $|\mathcal{V}|$ nodes and $|\mathcal{E}|$ edges. The feature matrix for the graph is denoted as $X \in \mathrm{R}^{|\mathcal{V}| \times d}$, where $i$-th row represents a $d$-dimensional feature vector of the $i$-th node $v_i$. $\mathbf{A} \in \{0,1\}^{n \times n}$ is the adjacency matrix where $\mathbf{A}_{i,j} = 1$ indi-

cates that there exists edge $e_{i,j} \in \mathcal{E}$ between node $v_i$ and $v_j$, and $\mathbf{A}_{i,j} = 0$ otherwise. In this paper, we assume that both ground-truth labels and sensitive attributes are binary variables for convenience. We let $\mathbf{S} \in \{0,1\}^{n \times 1}$ to denote the binary sensitive attributes, where $s_i$ is the sensitive attribute of $v_i$. We use $S_d = \{\forall\, v_i \in \mathcal{V} | s_i = 0\}$ denotes the deprived group (*e.g.*, female) and $S_f = \{\forall\, v_i \in \mathcal{V} | s_i = 1\}$ denotes the favored group (*e.g.*, male). For node classification, each node is also associated with a one-hot ground-truth node label $y_i$ where $\hat{y}_i$ is the label of $v_i$. We also assume $y_i = 1$ denotes the granted label and $y_i = 0$ denotes the rejected label. In addition, we let $\mathcal{G}_{v_i}$ represent the ego graph of center node $v_i$, which includes important neighbor nodes of the central node.

## 4. The Proposed Framework: FDGen

This section introduces FDGen, a framework built upon the graph diffusion model (Kong et al., 2023) to mitigate both structural and feature bias for fair graph generation. Given the correlation between these biases, FDGen is developed to adaptively mitigate them during generation, leveraging the flexibility and tractable probability distributions of diffusion models. Specifically, bias types in graph generation are first defined, along with theoretical foundations for fairness regularizers (Section 4.1). Next, we describe how FDGen isolates sensitive information into independent components to measure generation discrepancies and apply fairness constraints (Section 4.2). Finally, the FDGen model and its training objectives are presented (Section 4.3).

### 4.1. Fair Graph Generation Regularizer

We first examine the root causes of biases in graph generation, laying the groundwork for a fairness regularization

strategy to mitigate them. Specifically, graph generation models are vulnerable to bias through two main mechanisms: i) nodes with the same sensitive attributes tend to form denser connections, creating structural bias, and ii) the node feature distributions often show systematic differences across demographic groups, leading to feature bias. In addition, these biases can be mutually reinforcing, as nodes with similar features are more likely to connect, and connected nodes influence each other's feature representations. Specific to graph diffusion models, these biases can be further amplified during generation. Specifically, a diffusion-based graph generation model defines a forward Markov transition kernel $q(x_t \mid x_{t-1})$ to gradually corrupt the training data into a simple noise distribution. The model then learns a reverse denoising transition $p_\theta(x_{t-1} \mid x_t)$ using a neural network. During this reverse process, GNN-style message passing aggregates information from neighboring nodes to preserve both topological structure and node features. However, this message-passing mechanism can exacerbate existing biases through two effects: First, by smoothing representations of connected nodes, it strengthens segregation between different sensitive groups, amplifying graph structural bias. Second, it reinforces the association between node representations and sensitive attributes, exacerbating feature bias in the generated graphs.

To this end, a theoretically grounded fairness regularizer is proposed to mitigate both structural and feature bias simultaneously. Starting with node representation differences ($\mathbf{h}_D^l$), the commonly used approach is to measure them between groups as $\mathbf{h}_D^l = \mathbf{h}_{S_d}^l - \mathbf{h}_{S_f}^l$ ($\mathbf{h}_{S_d}^l$ and $\mathbf{h}_{S_f}^l$ denote the node representations for deprived and favored groups, respectively). However, while this captures overall representation differences between subgroups, it fails to account for the distinct roles of sensitive information, as nodes with different sensitive attributes should have different sensitive-related components while maintaining similar sensitive-irrelevant components. To this end, we separate node representations into sensitive-related representations $\mathbf{h}_i^{l,S}$ and sensitive-irrelevant representations $\mathbf{h}_i^{l,\overline{S}}$. This separation allows us to minimize the differences in sensitive-irrelevant components ($\mathbf{h}_i^{l,\overline{S}}$) while maintaining appropriate differences in sensitive-related components ($\mathbf{h}_i^{l,S}$), thereby preserving group identity while mitigating bias, as detailed in Definition 4.1.

**Definition 4.1** (Sensitive-irrelated Representation Discrepancy). Let $V_{S_d}$ and $V_{S_f}$ be two subgroups corresponding to distinct values $S_d, S_f \in \mathbf{S}$ of a sensitive attribute, with $S_d \neq S_f$. Suppose each node $v_i$ has a non-sensitive representation $\mathbf{h}_i^{l,\overline{S}}$ at layer $l$, intended to be independent of the sensitive attribute $S$. We quantify any unintended discrepancy in this non-sensitive space via the Maximum Mean Discrepancy (MMD):

$$\mathbf{h}_{\overline{\mathbf{S}}D}^{(l)} = \frac{1}{|V_{S_d}|^2} \sum_{v_i, v_j \in V_{S_d}} k\left(\mathbf{h}_{\overline{\mathbf{S}}i}^{(l)}, \mathbf{h}_{\overline{\mathbf{S}}j}^{(l)}\right) + \frac{1}{|V_{S_f}|^2} \sum_{v_i, v_j \in V_{S_f}} k\left(\mathbf{h}_{\overline{\mathbf{S}}i}^{(l)}, \mathbf{h}_{\overline{\mathbf{S}}j}^{(l)}\right)$$
$$- \frac{2}{|V_{S_d}| \cdot |V_{S_f}|} \sum_{\substack{v_i \in V_{S_d} \\ v_j \in V_{S_d}}} k\left(\mathbf{h}_{\overline{\mathbf{S}}i}^{(l)}, \mathbf{h}_{\overline{\mathbf{S}}j}^{(l)}\right)$$

where $k(\cdot, \cdot)$ is a kernel function (*e.g.*, RBF kernel). A large $\mathbf{h}_{\overline{\mathbf{S}}D}^{(l)}$ indicates substantial divergence between $V_{S_0}$ and $V_{S_1}$ the non-sensitive channel (*i.e.*, sub node representation). Conversely, a small MMD implies a better alignment of non-sensitive embeddings, reducing unfair discrimination risk in downstream tasks. In addition, this measure can be simplified to a direct Euclidean distance between mean embeddings, and extends naturally to multiple sensitive attributes.

Building on this representation separation measurement, FDGen proceeds in three steps. First, we analyze how representation discrepancy emerges during GNN-based aggregation, establishing an upper bound to understand its propagation through network layers (Theorem 4.2). Second, we demonstrate how this representation discrepancy directly influences group disparity, showing that reducing representation differences leads to improved demographic parity (Theorem 4.3). Finally, guided by these theoretical insights, we propose a fairness regularizer that simultaneously addresses both structural and feature bias by aligning non-sensitive representations while maintaining appropriate group-specific differences.

We begin with the first step of analyzing representation discrepancy. During GNN message-passing, node representations are influenced by both their neighbors and the network structure. Theorem 4.2 quantifies this influence by providing an upper bound on the representation discrepancy between sensitive groups, revealing how bias can accumulate across layers (proof in Appendix).

**Theorem 4.2.** *The discrepancy $\mathbf{h}_{\overline{\mathbf{S}}D}^{(l)}$ between the representations of nodes in a sensitive group $\mathcal{S}_d$ and the rest of the nodes at the $l^{th}$ GNN layer can be upper bounded by:*

$$\mathbf{h}_{\overline{\mathbf{S}}D}^{(l)} \leq \left(3 - \left(\frac{1}{|V_{S_d}||V_{S_f}|^2} + \frac{1}{|V_{S_d}|^2|V_{S_f}|}\right) \right.$$
$$\sum_{i \in \mathcal{S}_d, j \in \mathcal{S}_f} k\left(\mathbf{h}_{\overline{\mathbf{S}}i}^{l-1}, \mathbf{h}_{\overline{\mathbf{S}}j}^{l-1}\right) \left\|\mu_{l-1}^{(d)} - \mu_{l-1}^{(f)}\right\|$$
$$+ \left\|\mu^{(d)} - \mu^{(f)}\right\|$$
$$+ \left[2\sqrt{N}\|\mathbf{W}^{(l)}\|_\infty \left(\Delta^l + \|\boldsymbol{\mu}_l^{(d)} - \boldsymbol{\mu}_l^{(f)}\|\right)\right]\right) \tag{1}$$

where $\mu$ denotes the mean representation of a subgroup of nodes.

Building on Theorem 4.2, we next analyze how representation disparity influences group fairness. Theorem 4.3

demonstrates that by minimizing sensitive-irrelevant representation discrepancy, we can effectively reduce group disparity in model predictions (detailed proof in Appendix).

**Theorem 4.3.** *For a classification task, minimizing the sensitive representation discrepancy between two sensitive groups upper-bounds the group disparities:*

$$
\begin{aligned}
\Delta_{DP} &= \left| \frac{1}{|V_{S_d}|} \sum_{i \in \mathcal{S}_d} f(\mathbf{z}_i)_1 - \frac{1}{|V_{S_f}|} \sum_{j \in \mathcal{S}_f} f(\mathbf{z}_j)_1 \right| \\
&\leq \left| f(\mathbf{z}_{\mu^{(d)}})_1 - f(\mathbf{z}_{\mu^{(f)}})_1 \right| \\
&+ \frac{L}{2} \left( \frac{1}{|V_{S_d}|} \sum_{i=1}^{|V_{S_d}|} \|\mathbf{W}^{(l)}\| \|\mathbf{h}_{\overline{\mathbf{S}}D}^{(l)} + \frac{1}{|V_{S_f}|} \sum_{j=1}^{|V_{S_f}|} \|\mathbf{W}^{(l)}\| \|\mathbf{h}_{\overline{\mathbf{S}}D}^{(l)} \right)
\end{aligned}
\tag{2}
$$

Based on these theoretical results, we propose a fair regularizer $\mathcal{L}_T$ that reduces MMD in sensitive-irrelevant node representations while adjusting inter- and intra-group connections:

$$
\begin{aligned}
\mathcal{L}_T &= a\,\mathcal{L}_f + b\,\mathcal{L}_g \\
&= a\Big( \frac{1}{|V_{S_d}|^2} \sum_{v_i,v_j \in V_{S_d}} k(\mathbf{h}_i^{(l),\overline{S}}, \mathbf{h}_j^{(l),\overline{S}}) \\
&+ \frac{1}{|V_{S_f}|^2} \sum_{v_i,v_j \in V_{S_f}} k(\mathbf{h}_i^{(l),\overline{S}}, \mathbf{h}_j^{(l),\overline{S}}) \\
&- \frac{2}{|V_{S_d}|\cdot|V_{S_f}|} \sum_{v_i \in V_{S_d}, v_j \in V_{S_f}} k(\mathbf{h}_i^{(l),\overline{S}}, \mathbf{h}_j^{(l),\overline{S}}) \Big) \\
&+ b\left( \|\mathbf{E}_{\text{inter}} - \mathbf{A}_{\text{inter}}\|_F^2 + \|\mathbf{E}_{\text{intra}} - \mathbf{A}_{\text{intra}}\|_F^2 \right)
\end{aligned}
\tag{3}
$$

where $\mathcal{L}_f$ denotes the node-level fairness term that aligns $\mathbf{h}_{\overline{S}}^{(l)}$ across sensitive subgroups, and $\mathcal{L}_g$ captures the difference in modeling performance between inter- and intra-group edges, with $a$ and $b$ as hyperparameters that balance their contributions. In addition, $\mathbf{E}_{\text{inter}}$ and $\mathbf{E}_{\text{intra}}$ denote predicted inter- and intra-group connections, while $\mathbf{A}_{\text{inter}}$ and $\mathbf{A}_{\text{intra}}$ indicate their ground-truth adjacency.

In summary, $\mathcal{L}_T$ aims to align non-sensitive representations across different groups, while pushing the predicted graph edges to better match intra-group and inter-group statistics. $\mathcal{L}_T$, integrated into FDGen, thus functions to mitigate both structural and feature biases in the generated graphs.

### 4.2. Identifying Sensitive-irrelevant Representations

Equipped with the proposed fairness regularizer (Section 4.1), we now present how to disentangle node representations so as to isolate the components independent of sensitive attributes, thus enabling us to measure graph generation discrepancies and enforce fair graph generation regularizers. This process involves two main tasks: i) Decomposing each node representation into multiple components, where each component corresponds to a latent factor (*e.g.*, "age", "occupation"), and ii) Identifying which components capture sensitive attribute-related information and which are largely free from it.

In the first task, we aim to decompose the node embedding into multiple components, each representing a latent factor. When performing this task, using all neighbors to reconstruct the node component should be avoided, as only a subset carries useful information for the specific channel. For instance, if the latent factor is "family", neighbors denoting close family members should have higher weight, while unrelated neighbors should be down-weighted. To achieve such selective neighbor usage, we extend GAT-like attention into a multi-channel setting via a neighbor-assigner mechanism. Specifically, we assume that if two nodes $v_i$ and $v_j$ exhibit higher similarity in the $c^{th}$ component space, then factor $c$ is more likely to be responsible for their connection. For each channel $c \in \{1, 2, \ldots, N_c\}$, we compute the edge weight ($\omega_{v_i,v_j}^c$) between nodes $v_i$ and $v_j$ using dot-product attention to measure their similarity in channel $c$:

$$
\omega_{v_i,v_j}^c = \frac{\exp((h_{v_i}^c)^T h_{v_j}^c)}{\sum_{v_j \in \mathcal{N}(v_i)} \exp((h_{v_i}^c)^T h_{v_j}^c)}
\tag{4}
$$

where $\mathcal{N}(v_i)$ denote the neighbors of $v_i$. Using these edge weights ($\omega_{v_i,v_j}^c$), we adaptively aggregate information from relevant neighbors for each component. The aggregation process to obtain $h_{v_i,c}^{l+1}$ (the $c^{th}$ channel representation of node $v_i$ at layer $l+1$) is defined as follows:

$$
h_{v_i,c}^{l+1} = \sigma\Big( \sum_{(v_j) \in \hat{N}(v_i)} \omega_{(v_i,v_j)}^c \, \phi\big(h_{v_j,c}^l, \, h_{v_i,c}^l\big) \Big)
\tag{5}
$$

where $\sigma(\cdot)$ is a non-linear activation. By focusing on relevant neighbors in each channel, the aggregation in Equation (5) encourages $h_{v_i,c}^{l+1}$ to cluster nodes that share similar characteristics *under factor c*.

However, although the above mechanism decomposes the node embedding into multiple channels, it does not guarantee that these channels themselves are mutually independent. For instance, channel $c_1 =$ "age" might still be correlated with channel $c_2 =$ "nursing home". To address this, a disentangled constraint which adopts a distance covariance regularizer (Matteson & Tsay, 2017) that penalizes inter-channel correlations is proposed as follows:

$$
\mathcal{L}_{\text{d}} = \sum_{c_1=1}^{N_c} \sum_{c_2=c_1+1}^{N_c} \frac{d\mathrm{Cov}^2(Z^{c_1}, Z^{c_2})}{\sqrt{d\mathrm{Cov}^2(Z^{c_1}, Z^{c_1})\, d\mathrm{Cov}^2(Z^{c_2}, Z^{c_2})}}
\tag{6}
$$

where $N_c$ denote the number of channels and $Z^k$ denotes the set of node embeddings in channel $c$ (e.g., $Z^{c_1} = \{h_{(c_1,1)}, h_{(c_1,2)}, \ldots, h_{(c_1,N_{c_1})}\}$) and $cov(\cdot)$ is the distance covariance.

With the decomposed channels, the second task identifies which channel captures sensitive attribute information. Theorem 4.4 demonstrates that in fully disentangled representations, only one channel can be associated with each sensitive attribute.

**Theorem 4.4.** *Suppose we have $m$ channel representations $\{h_{c_1}, h_{c_2}, \ldots, h_{c_{N_c}}\}$ that are fully disentangled, meaning each channel is strictly independent from every other. Formally, $I\big(\mathbf{h}^{c_i}; \mathbf{h}^{c_j}\big) = 0, \forall i \neq j$. Then, at most one channel can capture the sensitive attribute.*

Hence, once we identify which channel is sensitive-related, the remaining channels can be treated as sensitive-attribute-independent. To this end, we train a channel-level discriminator that tries to predict a node's sensitive label $y_{s_i}$ from each channel embedding. Specifically, for channel $k$, we feed $h_{v_i}^k$ into a classifier to produce the predicted probability $\hat{y}_{s_i,k}$ of the sensitive label, and define the classification loss as follows:

$$\mathcal{L}_D = -\frac{1}{|\mathcal{V}_L|} \sum_{v_i \in \mathcal{V}_L} \sum_{c=1}^{N_c} \big[ y_{s_i} \log\big(\hat{y}_{s_i,c}\big) + \big(1 - y_{s_i}\big) \log\big(1 - \hat{y}_{s_i,c}\big) \big] \tag{7}$$

where $y_{s_i}$ is the ground truth sensitive attribute label for node $v_i$, and $\hat{y}_{s_i,c}$ is the prediction sensitive attribute. Channels that yield high classification accuracy for $y_{s_i}$ are deemed sensitive-related, while those with low accuracy are considered non-sensitive. Consequently, the fairness regularizer (Section 4.1) can be applied specifically to the sensitive-irrelevant components, ensuring minimal distributional discrepancy across different groups in the generated graphs.

### 4.3. Fair Graph Generative Process

This section presents the proposed fair graph generative model, FDGen, which incorporates both the fairness regularization constraint and the disentangled conditioning mechanism. The overall process is divided into two stages: i) the forward diffusion process, where the graph is gradually corrupted to obtain noisy samples, and ii) the reverse diffusion process, where we reconstruct the graph in a fairness-aware and disentangled manner. We will then describe each part in detail:

**Forward diffusion process.** During the forward diffusion process, FDGen constructs noisy versions of the input graph by successively absorbing (masking) individual nodes and their edges according to a Markov chain. Specifically, rather

than merely masking edges, FDGen learns a node ordering network that, at each diffusion step $t$, selects a node $v_i$ to absorb. In an autoregressive fashion, the node decay ordering $\sigma$ is sampled from $q_\phi(\sigma \mid G_0 = \{X_0, A_0\})$, where $q_\phi(\cdot \mid G_0)$ is a conditional probability distribution over possible node orderings $\sigma$. Here, $X_0$ and $A_0$ denote the initial node features and graph structure at time step 0. At subsequent time steps, $t$, $X_t$, and $A_t$ represent the node features and graph structure after absorbing one node. This process continues iteratively until the entire graph is absorbed. To systematically select which nodes to absorb at each step, the diffusion ordering network follows a recurrent structure:

$$q_\phi(\sigma \mid G_0) = \prod_t q_\phi\big(\sigma_t \mid G_0, \sigma_{(<t)}\big) \tag{8}$$

At each step $t$, the probability of selecting $\sigma_t$ depends on the original graph $G_0$ and all previously chosen nodes $\sigma_{(<t)}$. We employ a GNN to encode graph structure, incorporating positional encodings to represent the partial ordering. After running the GNN, we obtain an updated embedding $\mathbf{h}_i^d$ for each node $v_i$. The probability of selecting the next node $q_\phi(\sigma_t \mid G_0, \sigma_{(<t)})$ is then computed using a softmax over node embeddings:

$$q_\phi(\sigma_t \mid G_0, \sigma_{(<t)}) = \frac{\exp\big(\mathbf{h}_i^d\big)}{\sum_{i' \notin \sigma_{(<t)}} \exp\big(\mathbf{h}_{i'}^d\big)} \tag{9}$$

Through this approach, FDGen learns an ordering strategy that efficiently masks nodes during forward diffusion.

**Reverse diffusion process:** During the reverse diffusion process, our goal is to reconstruct both the node features $\mathbf{X}_0$ and the adjacency $\mathbf{A}_0$ from their noisy versions $\mathbf{X}_t$ and $\mathbf{A}_t$. Specifically, let $G_{0:n}$ represent all the intermediate states of the graph from $G_0$ to $G_n$, and let $q_\phi(\sigma_{1:n} \mid G_0)$ be the forward node-decay ordering network. To this end, FDGen uses maximum likelihood estimation, which optimizes the denoising model while guiding the denoising order through variational inference, leading to higher-quality generated graphs. The variational lower bound (VLB) of the likelihood of $G_0$ is:

$$\begin{aligned} \log p_\theta(G_0) &= \log\Big(\int p(G_{0:n}) \frac{q(G_{1:n} \mid G_0)}{q(G_{1:n} \mid G_0)} \, dG_{1:n}\Big) \\ &\geq \mathbb{E}_{q_\phi(\sigma_{1:n} \mid G_0)} \sum_t \log p_\theta\big(G_t \mid G_{t+1}\big) \\ &\quad - \mathrm{KL}\Big(q_\phi(\sigma_{1:n} \mid G_0) \,\big\|\, p_{\theta'}(\sigma_{1:n} \mid G_n)\Big) \end{aligned} \tag{10}$$

Due to node permutation invariance (Kong et al., 2023), there is no need to learn a separate reverse node-generation ordering, allowing the omission of the KL term in the variational lower bound and leading to a simplified objective. The

training objective ($\mathcal{L}_P$), which is minimized using stochastic gradient descent, is thus defined as:

$$
\begin{aligned}
\mathcal{L}_{\mathrm{P}} &= -\mathbb{E}_{\sigma_{1:n} \sim q_\phi(\sigma_{1:n}|G_0)} \sum_t \log p_\theta\big(G_t \mid G_{t+1}\big) \\
&= -n\, \mathbb{E}_{\sigma_{1:n} \sim q_\phi(\sigma_{1:n}|G_0)} \mathbb{E}_{t \sim \mathcal{U}_n} \log p_\theta\big(O_{\sigma_t}^{\pi(>t)} \mid G_{t+1}\big)
\end{aligned}
\tag{11}
$$

where $t$ is drawn uniformly from $\{1, \ldots, n\}$ and $O_{\sigma_t}^{\pi(>t)}$ indicates the node $v_{\sigma_t}$ and its edges with any previously unmasked nodes $\{\sigma_{t+1}, \ldots, \sigma_n\}$.

While Equation 11 guides the model to learn a generative process that reconstructs node features and edges, it does not explicitly address bias nor disentanglement. To ensure fairness across different groups and decompose node representations into multiple channels (Section 4.2), for a node $v_i$, the node embedding at the $l^{th}$ layer follows Equation 5.

Building on this, we integrate the proposed fairness constraint into the denoising network $p_\theta(G_t \mid G_{t+1})$. Specifically, we use $\mathcal{L}_f$ to reduce feature discrepancies among different sensitive groups and $\mathcal{L}_g$ to constrain inter-group and intra-group edge biases. Hence, we augment the likelihood term in Equation 11 with these fairness and disentanglement losses. During each reverse diffusion step, the aggregator (Equation 5) updates node embeddings split by channel, and we measure fairness and disentanglement on the resulting representations. Thus, the final training objective is:

$$
\mathcal{L}_{\mathrm{total}} = \mathcal{L}_P + \lambda_1 \mathcal{L}_f + \lambda_2 \mathcal{L}_g + \lambda_3 \mathcal{L}_d + \lambda_4 \mathcal{L}_D \tag{12}
$$

where $\mathcal{L}_{\mathrm{train}}$ ensures node and edge reconstruction, $\mathcal{L}_f$ and $\mathcal{L}_g$ mitigate bias in node features and topology, and $\mathcal{L}_d$ with $\mathcal{L}_D$ enforces disentangled channels and isolates the sensitive attribute. Note that FDGen preserves $S$ throughout training and sampling (*i.e.*, $S$ is initialized according to its original distribution at inference time) so that FDGen can capture how sensitive attributes correlate with graph structure without amplifying undesirable biases.

# 5. Experiment

**Datasets.** Four real-world fairness datasets, namely Cora, Citeseer, Photo, and Computer, are used in our experiments. We provide a short overview of these datasets as follows: In the Cora and Citeseer datasets (Sen et al., 2008), nodes represent papers, edges capture citation relationships between papers, node features are bag-of-words vectors of keywords, and labels indicate the research field of the papers. The Photo and Computer datasets (Shchur et al., 2018) are segments of the Amazon co-purchase graph, where nodes represent products, edges show that two products are frequently

purchased together, node features are bag-of-words vectors derived from product reviews, and labels correspond to the product category. Table 1 summarizes the statistics of these datasets.

*Table 1.* Summary of the datasets used in the experiments.

| Dataset | Cora | Citeseer | Photo | Computer |
|---|---|---|---|---|
| # Nodes | 2,708 | 3,327 | 7,650 | 13,752 |
| # Edges | 10,556 | 9,228 | 238,163 | 491,722 |
| # Features | 1,433 | 3,703 | 745 | 767 |
| # Average Degree | 3.89 | 2.77 | 31.13 | 35.75 |
| Sensitive Attribute | Topic | Topic | Product Categories | Product Categories |

**Baselines.** We compare the proposed method with the following baselines: GRAPHARM (Kong et al., 2023), FairAdj (Li et al., 2021), FG$^2$AN (Wang et al., 2023c), FairGen (Zheng et al., 2024) and FairWire (Kose & Shen, 2024b).

**Evaluation Metrics.** We evaluate FDGen's performance across two aspects: 1) Node Classification Performance: We use accuracy and F1 scores to evaluate node classification utility, along with $\Delta$DP (Dwork et al., 2012) and $\Delta$EO (Hardt et al., 2016) to assess prediction fairness. 2) Generated Graph Quality: Following (Kong et al., 2023), we measure generation quality using maximum mean discrepancy (MMD) between generated and input graphs, specifically computing MMD for degree distribution and clustering coefficient. To evaluate graph structural fairness, we propose metrics, fair degree distribution and fair clustering coefficient, that measure disparity between subgroups. These are computed as: $f(\mathcal{G}_{S_d}, \tilde{\mathcal{G}}_{S_d}) - f(\mathcal{G}_{S_f}, \tilde{\mathcal{G}}_{S_f})$, where $f(\cdot)$ denotes the MMD value for degree distribution or clustering coefficient, and $\tilde{\mathcal{G}}$ represents the generated graph.

## 5.1. Experiment Results

**Graph Generation Results.** We evaluate our generative models in terms of both quality and fairness, with results shown in Figure 2. FDGen demonstrates strong performance in graph generation, achieving comparable or better quality than baseline methods while significantly improving fairness metrics. The strong performance can be attributed to two key factors. First, FDGen maintains group identity information through decomposition learning while reducing differences in sensitive-irrelevant attributes. This approach enables FDGen to achieve fairness with minimal compromise to synthetic graph quality compared to other fairness-aware baselines. Second, by simultaneously addressing both structural and feature bias, FDGen better mitigates group differences in generated graphs compared to existing methods that focus solely on structural bias. Overall, these experi-

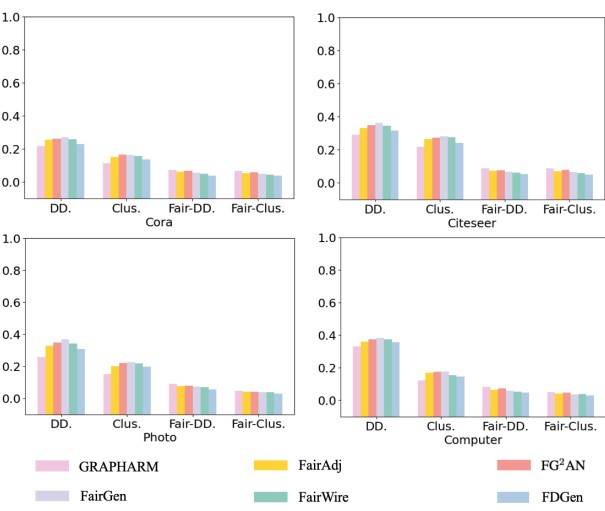

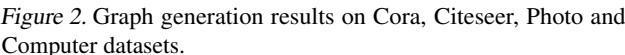

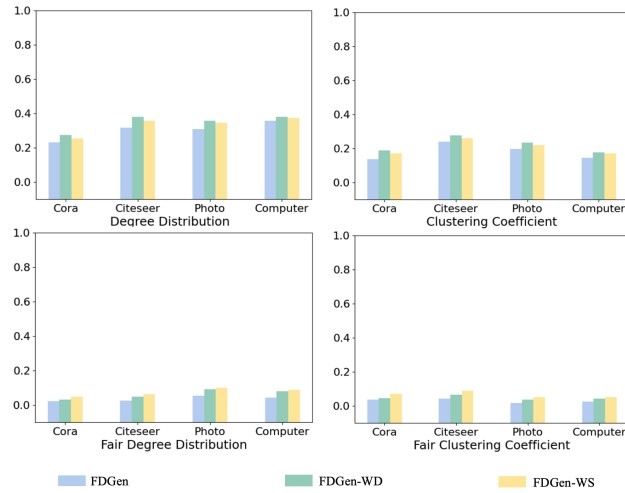

*Figure 2.* Graph generation results on Cora, Citeseer, Photo and Computer datasets.

*Figure 3.* Ablation study results for FDGen, FDGen-WD, and FDGen-WS.

mental results demonstrate that FDGen effectively improves fairness while maintaining graph generation quality.

*Table 2.* Node classification results on the Cora dataset.

| Cora dataset | ACC (%) | $\Delta_{DP}(\%)$ | $\Delta_{EO}(\%)$ |
|---|---|---|---|
| Original-GCN | **82.43 ± 0.34** | 27.01 ± 1.38 | 25.21 ± 1.13 |
| GRAPHARM-GCN | 81.03 ± 0.23 | 25.21 ± 1.38 | 21.31 ± 1.43 |
| FairAdj-GCN | 77.77 ± 1.64 | 17.13 ± 6.36 | 13.96 ± 2.24 |
| FG$^2$AN-GCN | 78.10 ± 0.81 | 18.66 ± 4.30 | 14.05 ± 0.32 |
| FairGen-GCN | 79.54 ± 1.56 | 14.16 ± 0.89 | 13.35 ± 1.24 |
| FairWire-GCN | 78.21 ± 1.03 | 14.76 ± 0.24 | 13.65 ± 0.51 |
| **FDGen-GCN** | 80.05 ± 1.03 | **13.88 ± 0.24** | **11.95 ± 0.37** |

**Node Classification Results.** We evaluate both the accuracy and fairness of node classification using generated graphs. For this evaluation, we use GCN as our base model, training it on generated graphs and testing its predictive performance. Using the Cora dataset as a case study, Table 2 shows the comparison between FDGen and baseline methods on the node classification task. The results demonstrate that graphs generated by FDGen consistently lead to better classifier performance in terms of both accuracy and fairness. This improvement stems from FDGen's ability to mitigate both structural and feature bias during graph generation, thus reducing bias propagation from raw graph data to downstream tasks.

**Ablation Study.** To verify the effectiveness of our proposed modules, we conduct an ablation study examining two variants of FDGen. First, we analyze the impact of identifying sensitive-irrelevant representations by creating FDGen-WD, which applies fair regularization directly to complete node representations. As shown in Figure 3, FDGen-WD shows

reduced graph generation quality compared to the complete FDGen model. This decline in performance occurs because eliminating group identity information reduces the authenticity of generated graphs.

Second, we study the effect of mitigating only feature bias by creating FDGen-WS. Results in Figure 3 show that FDGen-WS achieves worse fairness compared to the complete FDGen model. While nodes may have similar sensitive-irrelevant attributes, their connections remain influenced by sensitive-related information, leading to increased connectivity between nodes sharing sensitive attributes and introducing structural bias. However, FDGen-WS maintains better graph generation quality than FDGen-WD. Overall, these findings underscore the necessity of our design choices.

## 6. Conclusion

Generating synthetic graphs that capture structural characteristics of real data has gained increasing attention as a solution to scalability and privacy challenges in real-world networks. However, fairness in graph generation models remains an important yet understudied problem. This work addresses this gap by investigating bias in graph data and proposing FDGen, a framework that mitigates both structural and feature bias. Our theoretically grounded fairness regularizer effectively reduces identified bias factors, as demonstrated through extensive experiments comparing FDGen with both fairness-agnostic and fairness-aware baselines on real and synthetic graphs. These results establish a foundation for developing fair graph generation models and open possibilities for future work on a comprehensive study of graph learning.

## Acknowledgements

This work was supported in part by the National Science Foundation (NSF) under Grant No. 2404039.

## Impact Statement

Graph generation models are increasingly used in real-world applications spanning from social networks to financial systems. When these models inherit and amplify societal biases, they can perpetuate discrimination and unfair treatment of disadvantaged groups. Our work advances fairness in artificial intelligence by developing methods to identify and mitigate both structural and feature bias in generated graphs. This research has particular significance for high-stakes applications such as credit scoring and healthcare, where biased graph generation could disproportionately affect vulnerable populations. By enabling fairer graph generation while maintaining data utility, our work contributes to more equitable AI systems and provides a foundation for future research on structural fairness in graph learning.

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

## A. Proof of Theorem 4.2

To simplify the notation, we use $\mathbf{h}^l$ to denote the input representations, and $\mathbf{h}^{l+1}$ to denote the output representations. The considered disparity measure follows as:

$$\mathbf{h}_{\overline{\mathbf{S}}D}^{(l)} = \mathrm{MMD}\Big(\{\mathbf{hs}_i^l \mid v_i \in V_{S_d}\}, \{\mathbf{hs}_i^l \mid v_i \in V_{S_f}\}\Big) \tag{13}$$

Expand under the RBF kernel function: $k(x,y) = \exp\left(-\gamma\|x-y\|^2\right)$, we can get:

$$\mathbf{h}_{\overline{\mathbf{S}}D}^{(l)} = \frac{1}{|V_{S_d}|^2} \sum_{v_i,v_j \in V_{S_d}} k\Big(\mathbf{h}_{\overline{S}i}^l, \mathbf{h}_{\overline{S}j}^l\Big) + \frac{1}{|V_{S_f}|^2} \sum_{v_i,v_j \in V_{S_f}} k\Big(\mathbf{h}_{\overline{S}i}^l, \mathbf{h}_{\overline{S}j}^l\Big) - \frac{2}{|V_{S_d}| \cdot |V_{S_f}|} \sum_{\substack{v_i \in V_{S_d} \\ v_j \in V_{S_f}}} k\Big(\mathbf{h}_{\overline{S}i}^l, \mathbf{h}_{\overline{S}j}^l\Big) \tag{14}$$

Building on this, and given that Graph Attention Networks (GAT) adopt the message passing by assigning different weights to neighbor nodes as:

$$\mathbf{h}_i^{(l)} = \sum_{v_j \in \mathcal{N}(i)} a_{ij}^{(l-1)} \mathbf{h}_j^{(l-1)} \quad \text{with} \quad a_{ij}^{(l-1)} = \frac{\exp\left(e_{ij}^{(l-1)}\right)}{\sum_{v_j \in \mathcal{N}(i)} \exp\left(e_{ij}^{(l-1)}\right)} \tag{15}$$

Hence, we can re-write the disparity as follows:

$$\begin{aligned}
\mathbf{h}_i^{(l)} = \mathbf{X}_i \;&+\; \sum_{u \in \mathcal{N}_i \cap S_d} a_{iu}^{(l-1)} \mathbf{h}_u^{(l-1)} \;+\; \sum_{u \in \mathcal{N}_i \cap S_f} a_{iu}^{(l-1)} \mathbf{h}_u^{(l-1)} \\
&- \frac{1}{N_d^2} \sum_{u \in S_d} k\big(\mathbf{h}_u^{(l-1)}, \mathbf{h}_i^{(l-1)}\big) \mathbf{h}_u^{(l-1)} \;+\; \frac{1}{N_d N_f} \sum_{u \in S_f} k\big(\mathbf{h}_u^{(l-1)}, \mathbf{h}_i^{(l-1)}\big) \mathbf{h}_u^{(l-1)} \\
&+ \Big( \frac{1}{N_d^2} \sum_{u \in S_d} k\big(\mathbf{h}_u^{(l-1)}, \mathbf{h}_i^{(l-1)}\big) \;-\; \frac{1}{N_d N_f} \sum_{u \in S_f} k\big(\mathbf{h}_u^{(l-1)}, \mathbf{h}_i^{(l-1)}\big) \Big) \mathbf{h}_i^{(l-1)}
\end{aligned} \tag{16}$$

Given that each node $u \in S_d$, the node representation $h_u^{(l)}$ subject to $\mu_i^{(d)} - \Delta^l \preceq h_u^{(l)} \preceq \mu_i^{(d)} + \Delta^l$ (Kose & Shen, 2024a), where the parameter $\Delta^l$ serves as a per-layer tolerance indicating how far the representation is allowed to deviate from $\mu_i^{(d)}$ along each coordinate. If $u$ instead belongs to $S_f$, a parallel condition applies with $\mu_i^{(f)} \pm \Delta^l$, anchoring $h_u^{(l)}$ around the group mean $\mu_i^{(f)}$. Building on this, we assume the node representation within a controlled region near its respective group's mean. Hence, we can define the $\mathbf{h}_i^{(l)}$ as:

$$\begin{aligned}
\mathbf{h}_i^{(l)} \in \Bigg[ &\mu^{(d)} + \sum_{u \in \mathcal{N}_i \cap \mathcal{S}_d} a_{iu}^{(l-1)} \mathbf{h}_u^{(l-1)} + \sum_{u \in \mathcal{N}_i \cap \mathcal{S}_f} a_{iu}^{(l-1)} \mathbf{h}_u^{(l-1)} \\
&- \frac{1}{N_d^2} \sum_{u \in \mathcal{S}_d} k\big(\mathbf{h}_u^{(l-1)}, \mathbf{h}_i^{(l-1)}\big) \mathbf{h}_u^{(l-1)} + \frac{1}{N_d N_f} \sum_{u \in \mathcal{S}_f} k\big(\mathbf{h}_u^{(l-1)}, \mathbf{h}_i^{(l-1)}\big) \mu_{l-1}^{(f)} \\
&+ \frac{1}{N_d^2} \sum_{u \in \mathcal{S}_d} k\big(\mathbf{h}_u^{(l-1)}, \mathbf{h}_i^{(l-1)}\big) \mathbf{h}_i^{(l-1)} - \frac{1}{N_d N_f} \sum_{u \in \mathcal{S}_f} k\big(\mathbf{h}_u^{(l-1)}, \mathbf{h}_i^{(l-1)}\big) \mu_{l-1}^{(d)} \\
&\pm \Big[ 2\sqrt{N} \|\mathbf{W}^{(l)}\|_\infty \Big( \Delta^l + \|\boldsymbol{\mu}_l^{(d)} - \boldsymbol{\mu}_l^{(f)}\| \Big) \Big] \Bigg]
\end{aligned}$$

Building on this, for nodes $v_i \in S_d$, we have:

$$\frac{1}{N_d} \sum_{i \in \mathcal{S}_d} \mathbf{h}_i^{(l)} \in \left[ \mu^{(d)} + \left( \mu_{l-1}^{(d)} + \frac{1}{N_d} \sum_{i \in \mathcal{S}_d} \alpha_i^{(l-1)} \left( \mu_{l-1}^{(f)} - \mu_{l-1}^{(d)} \right) \right) \right.$$
$$\left. + \frac{1}{N_d^2 N_f} \sum_{i \in \mathcal{S}_d} \sum_{u \in \mathcal{S}_f} k(\mathbf{h}_u^{(l-1)}, \mathbf{h}_i^{(l-1)}) \left( \mu_{l-1}^{(f)} - \mu_{l-1}^{(d)} \right) \pm \left[ 2\sqrt{N} \|\mathbf{W}^{(l)}\|_\infty \left( \Delta^l + \|\boldsymbol{\mu}_l^{(d)} - \boldsymbol{\mu}_l^{(f)}\| \right) \right] \right]$$

(17)

We can do similar for $S_f$.

$$\frac{1}{N_f} \sum_{j \in \mathcal{S}_f} \mathbf{h}_j^{(l)} \in \left[ \mu^{(f)} + \left( \mu_{l-1}^{(f)} + \frac{1}{N_f} \sum_{j \in \mathcal{S}_f} \alpha_j^{(l-1)} \left( \mu_{l-1}^{(d)} - \mu_{l-1}^{(f)} \right) \right) \right.$$
$$\left. + \frac{1}{N_d N_f^2} \sum_{j \in \mathcal{S}_f} \sum_{u \in \mathcal{S}_d} k(\mathbf{h}_u^{(l-1)}, \mathbf{h}_j^{(l-1)}) \left( \mu_{l-1}^{(d)} - \mu_{l-1}^{(f)} \right) \pm \left[ 2\sqrt{N} \|\mathbf{W}^{(l)}\|_\infty \left( \Delta^l + \|\boldsymbol{\mu}_l^{(d)} - \boldsymbol{\mu}_l^{(f)}\| \right) \right] \right]$$

(18)

We define the upper bound of the consequent representation discrepancy on node representation between two sensitive groups as follows:

$$\mathbf{h}_D^{(l)} = \left\| \tfrac{1}{N_d} \sum_{i \in \mathcal{S}_d} \mathbf{h}_i^{(l)} - \tfrac{1}{N_f} \sum_{j \in \mathcal{S}_f} \mathbf{h}_j^{(l)} \right\|$$
$$\leq \left| 1 - \left( \tfrac{1}{N_d} \sum_{i \in \mathcal{S}_d} \sum_{u \in \mathcal{N}_i \cap \mathcal{S}_{\mathrm{opp}}(v_i)} a_{iu}^{(l-1)} + \tfrac{1}{N_f} \sum_{j \in \mathcal{S}_f} \sum_{u \in \mathcal{N}_j \cap \mathcal{S}_{\mathrm{opp}}(v_j)} a_{ju}^{(l-1)} \right) \right.$$
$$\left. - \left( \tfrac{1}{N_d N_f^2} + \tfrac{1}{N_d^2 N_f} \right) \sum_{i \in \mathcal{S}_d} \sum_{j \in \mathcal{S}_f} k(\mathbf{h}_i^{(l-1)}, \mathbf{h}_j^{(l-1)}) \right| \left\| \mu_{l-1}^{(d)} - \mu_{l-1}^{(f)} \right\|$$
$$+ \left\| \mu^{(d)} - \mu^{(f)} \right\| + \left[ 2\sqrt{N} \|\mathbf{W}^{(l)}\|_\infty \left( \Delta^l + \|\boldsymbol{\mu}_l^{(d)} - \boldsymbol{\mu}_l^{(f)}\| \right) \right]$$
$$\leq \left( \tfrac{1}{N_d} \sum_{i \in \mathcal{S}_d} \sum_{u \in \mathcal{N}_i \cap \mathcal{S}_{\mathrm{opp}}(v_i)} a_{iu}^{(l-1)} + \tfrac{1}{N_f} \sum_{j \in \mathcal{S}_f} \sum_{u \in \mathcal{N}_j \cap \mathcal{S}_{\mathrm{opp}}(v_j)} a_{ju}^{(l-1)} \right) + 1$$

(19)

$$- \left( \tfrac{1}{N_d N_f^2} + \tfrac{1}{N_d^2 N_f} \right) \sum_{i \in \mathcal{S}_d} \sum_{j \in \mathcal{S}_f} k(\mathbf{h}_i^{(l-1)}, \mathbf{h}_j^{(l-1)}) \left\| \mu_{l-1}^{(d)} - \mu_{l-1}^{(f)} \right\|$$
$$+ \left\| \mu^{(d)} - \mu^{(f)} \right\| + \left[ 2\sqrt{N} \|\mathbf{W}^{(l)}\|_\infty \left( \Delta^l + \|\boldsymbol{\mu}_l^{(d)} - \boldsymbol{\mu}_l^{(f)}\| \right) \right]$$
$$\leq \left( 3 - \left( \tfrac{1}{N_d N_f^2} + \tfrac{1}{N_d^2 N_f} \right) \sum_{i \in \mathcal{S}_d} \sum_{j \in \mathcal{S}_f} k(\mathbf{h}_i^{(l-1)}, \mathbf{h}_j^{(l-1)}) \right) \left\| \mu_{l-1}^{(d)} - \mu_{l-1}^{(f)} \right\|$$
$$+ \left\| \mu^{(d)} - \mu^{(f)} \right\| + \left[ 2\sqrt{N} \|\mathbf{W}^{(l)}\|_\infty \left( \Delta^l + \|\boldsymbol{\mu}_l^{(d)} - \boldsymbol{\mu}_l^{(f)}\| \right) \right]$$

Noting that each node representation $\mathbf{h}_i$ can be decomposed as $\mathbf{h}_i = \mathbf{h}_{iS} \oplus \mathbf{h}_{\overline{S}i}$ (sensitive vs. non-sensitive part), we obtain the analogous upper bound for the sensitive-irrelevant representation $\mathbf{h}_{\overline{S},D}^{(l)}$. Hence, the representation discrepancy between different sensitive groups in the non-sensitive subspace is also bounded by:

$$\mathbf{h}_{\overline{S},D}^{(l)} \leq \left( 3 - \left( \frac{1}{|V_{S_d}| |V_{S_f}|^2} + \frac{1}{|V_{S_d}|^2 |V_{S_f}|} \right) \sum_{i \in \mathcal{S}_d, j \in \mathcal{S}_f} k(\mathbf{h}_{\overline{S},i}^{l-1}, \mathbf{h}_{\overline{S},j}^{l-1}) \right) \|\mu_{l-1}^{(d)} - \mu_{l-1}^{(f)}\| + \|\mu^{(d)} - \mu^{(f)}\|$$
$$+ \left[ 2\sqrt{N} \|\mathbf{W}^{(l)}\|_\infty \left( \Delta^l + \|\boldsymbol{\mu}_l^{(d)} - \boldsymbol{\mu}_l^{(f)}\| \right) \right]$$

(20)

which concludes the proof.

# B. Proof of Theorem 4.3

For binary classification, statistical parity is defined as: $\Delta_{SP} = |P(\hat{y} = 1|s = 0) - P(\hat{y} = 1|s = 1)|$. We consider the binary classification task and examine the properties of the Softmax function in this context. Let $P_1$ and $P_2$ represent the probabilities of class 1 ($c_1$) and class 2 ($c_2$), respectively. The function $Softmax(\cdot)$ is Lipschitz continuous with a Lipschitz constant $L$. Due to this Lipschitz continuity, the difference in output probabilities can be bounded by the difference in input vectors:

$$\begin{aligned} \|f(\mathbf{h_i}) - f(\mathbf{h_j})\| &= |P_1 - P_2| + |(1 - P_1) - (1 - P_2)| \\ &= 2|P_1 - P_2| \leq L\|\mathbf{h_i} - \mathbf{h_j}\| \end{aligned} \tag{21}$$

where $\mathbf{h_i}$ is the node representation for $\forall v_i \in S_d$ and $\mathbf{h_j}$ is the node representation for $\forall v_j \in S_f$.

Building on this, we can rewrite the statistical parity as follows:

$$\Delta_{\text{DP}} = \left| \frac{1}{N_d} \sum_{i \in \mathcal{S}_d} f(\mathbf{z}_i)_1 - \frac{1}{N_f} \sum_{j \in \mathcal{S}_f} f(\mathbf{z}_j)_1 \right| \tag{22}$$

where $\mathbf{z}_i = W^l \mathbf{h}_i^{(l)}$, and $W^{(l)}$ is the weight matrix at layer $l$.

As we discuss above, from Equation 21 and using node $v_i$ from the group $S_d$ as an example, we can get:

$$2\left| f(\mathbf{z}_i)_1 - f(\mathbf{z}_{\mu^{(d)}})_1 \right| \leq L\|\mathbf{z}_i - \mathbf{z}_{\mu^{(d)}}\| \tag{23}$$

Therefore, we can rewrite it as:

$$f(\mathbf{z}_{\mu^{(d)}})_1 - \tfrac{L}{2}\|\mathbf{z}_i - \mathbf{z}_{\mu^{(d)}}\| \leq f(\mathbf{z}_i)_1 \leq f(\mathbf{z}_{\mu^{(d)}})_1 + \tfrac{L}{2}\|\mathbf{z}_i - \mathbf{z}_{\mu^{(d)}}\| \tag{24}$$

Let $\mathbf{z}_i = \mathbf{W}^{(l)} \mathbf{h}_i^{(l)}$ for node $i$, and $\mathbf{z}_{\mu^{(d)}} = \mathbf{W}^{(l)} \mu_l^{(d)}$, $\mathbf{z}_{\mu^{(f)}} = \mathbf{W}^{(l)} \mu_l^{(f)}$ be the group means in logits space.

$$\begin{aligned} f(\mathbf{z}_{\mu^{(d)}})_1 - f(\mathbf{z}_{\mu^{(f)}})_1 - \tfrac{1}{N_d}\sum_{i=1}^{N_d} \tfrac{L}{2}\|\mathbf{z}_i - \mathbf{z}_{\mu^{(d)}}\| - \tfrac{1}{N_f}\sum_{j=1}^{N_f} \tfrac{L}{2}\|\mathbf{z}_j - \mathbf{z}_{\mu^{(f)}}\| \\ \leq \tfrac{1}{N_d}\sum_{i \in \mathcal{S}_d} f(\mathbf{z}_i)_1 - \tfrac{1}{N_f}\sum_{j \in \mathcal{S}_f} f(\mathbf{z}_j)_1 \leq \\ f(\mathbf{z}_{\mu^{(d)}})_1 - f(\mathbf{z}_{\mu^{(f)}})_1 + \tfrac{1}{N_d}\sum_{i=1}^{N_d} \tfrac{L}{2}\|\mathbf{z}_i - \mathbf{z}_{\mu^{(d)}}\| + \tfrac{1}{N_f}\sum_{j=1}^{N_f} \tfrac{L}{2}\|\mathbf{z}_j - \mathbf{z}_{\mu^{(f)}}\| \end{aligned}$$

Consider the $h_D$ we obtained:

$$\begin{aligned} \|\mathbf{z}_i - \mathbf{z}_{\mu^{(d)}}\| &= \|\mathbf{W}^{(l)}(\mathbf{h}_i^{(l)} - \mu_i^{(d)})\| \\ &\leq \|\mathbf{W}^{(l)}\| \left[ \left(3 - \left(\frac{1}{N_d N_f^2} + \frac{1}{N_d^2 N_f}\right) \sum_{p \in \mathcal{S}_d} \sum_{q \in \mathcal{S}_f} k(\mathbf{h}_p^{(l-1)}, \mathbf{h}_q^{(l-1)})\right) \|\mu_{l-1}^{(d)} - \mu_{l-1}^{(f)}\| \right. \\ &\left. + \|\mu^{(d)} - \mu^{(f)}\| + \left[2\sqrt{N}\left(\Delta^l + \|\boldsymbol{\mu}_l^{(d)} - \boldsymbol{\mu}_l^{(f)}\|\right)\right] \right] \end{aligned} \tag{25}$$

and we can get similiar for $\|\mathbf{z}_i - \mathbf{z}_{\mu^{(f)}}\|$.

Building on this, we re-express the upper bound.

$$
\begin{aligned}
\Delta_{\mathrm{DP}} &= \left| \frac{1}{N_d} \sum_{i \in \mathcal{S}_d} f(\mathbf{z}_i)_1 \; - \; \frac{1}{N_f} \sum_{j \in \mathcal{S}_f} f(\mathbf{z}_j)_1 \right| \\
&\leq \; \left[ f(\mathbf{z}_{\mu^{(d)}})_1 \; - \; f(\mathbf{z}_{\mu^{(f)}})_1 \right] \; + \; \frac{L}{2} \Big( \frac{1}{N_d} \sum_{i=1}^{N_d} \|\mathbf{z}_i - \mathbf{z}_{\mu^{(d)}}\| + \frac{1}{N_f} \sum_{j=1}^{N_f} \|\mathbf{z}_j - \mathbf{z}_{\mu^{(f)}}\| \Big)
\end{aligned}
\tag{26}
$$

Here we have combined the group center difference $\left[ f(\mathbf{z}_{\mu^{(d)}})_1 - f(\mathbf{z}_{\mu^{(f)}})_1 \right]$ with the Lipschitz offset $\frac{1}{N_d} \sum \|\mathbf{z}_i - \mathbf{z}_{\mu^{(d)}}\| + \frac{1}{N_f} \sum \|\mathbf{z}_j - \mathbf{z}_{\mu^{(f)}}\|$, factoring out $\frac{L}{2}$.

We note that

$$
\|\mathbf{z}_i - \mathbf{z}_{\mu^{(d)}}\| = \|\mathbf{W}^{(l)}(\mathbf{h}_i^{(l)} - \mu_l^{(d)})\| \; \leq \; \|\mathbf{W}^{(l)}\| \, \|\mathbf{h}_i^{(l)} - \mu_l^{(d)}\|
\tag{27}
$$

Similarly for $\mathbf{z}_j - \mathbf{z}_{\mu^{(f)}}$. According to (1), we know

$$
\begin{aligned}
\|\mathbf{h}_i^{(l)} - \mu_l^{(d)}\| &\leq \Big( 3 - \Big( \tfrac{1}{N_d \, N_f^2} + \tfrac{1}{N_d^2 \, N_f} \Big) \sum_{p \in \mathcal{S}_d} \sum_{q \in \mathcal{S}_f} k\big( \mathbf{h}_p^{(l-1)}, \mathbf{h}_q^{(l-1)} \big) \Big) \|\mu_{l-1}^{(d)} - \mu_{l-1}^{(f)}\| \\
&\quad + \|\mu^{(d)} - \mu^{(f)}\| + \left[ 2\sqrt{N} \|\mathbf{W}^{(l)}\|_\infty \Big( \Delta^l + \|\boldsymbol{\mu}_l^{(d)} - \boldsymbol{\mu}_l^{(f)}\| \Big) \right]
\end{aligned}
\tag{28}
$$

By Theorem 4.2, $\|\mathbf{h}_i^{(l)} - \mu_l^{(d)}\|$ is itself bounded by a term involving $\|\mu_{l-1}^{(d)} - \mu_{l-1}^{(f)}\|$, plus $L \|\boldsymbol{\Delta}^{(l-1)}\|$ and $C \|\Delta_z\|$. Thus,

$$
\begin{aligned}
\|\mathbf{z}_i - \mathbf{z}_{\mu^{(d)}}\| &\leq \|\mathbf{W}^{(l)}\| \left[ \Big( 3 - \Big( \tfrac{1}{N_d \, N_f^2} + \tfrac{1}{N_d^2 \, N_f} \Big) \sum_{p \in \mathcal{S}_d} \sum_{q \in \mathcal{S}_f} k\big( \mathbf{h}_p^{(l-1)}, \mathbf{h}_q^{(l-1)} \big) \Big) \|\mu_{l-1}^{(d)} - \mu_{l-1}^{(f)}\| \right. \\
&\quad \left. + \|\mu^{(d)} - \mu^{(f)}\| + \left[ 2\sqrt{N} \Big( \Delta^l + \|\boldsymbol{\mu}_l^{(d)} - \boldsymbol{\mu}_l^{(f)}\| \Big) \right] \right]
\end{aligned}
\tag{29}
$$

Similarly for $\|\mathbf{z}_j - \mathbf{z}_{\mu^{(f)}}\|$. Plugging these bounds into (24) and summing over $v_i \in S_d, v_j \in S_f$ yields:

$$
\begin{aligned}
\Delta_{\mathrm{DP}} \; \leq \; & \left| f(\mathbf{z}_{\mu^{(d)}})_1 - f(\mathbf{z}_{\mu^{(f)}})_1 \right| \\
& + \frac{L}{2} \|\mathbf{W}^{(l)}\| \left[ \frac{1}{N_d} \sum_{i=1}^{N_d} \Big( \Big( 3 - \Big( \tfrac{1}{N_d \, N_f^2} + \tfrac{1}{N_d^2 \, N_f} \Big) \right. \\
& \qquad\qquad \sum_{p,q} k(\mathbf{h}_p^{(l-1)}, \mathbf{h}_q^{(l-1)}) \Big) \|\mu_{l-1}^{(d)} - \mu_{l-1}^{(f)}\| \\
& \qquad + \|\mu^{(d)} - \mu^{(f)}\| + \left[ 2\sqrt{N} \Big( \Delta^l + \|\boldsymbol{\mu}_l^{(d)} - \boldsymbol{\mu}_l^{(f)}\| \Big) \right] \Big) \\
& + \frac{1}{N_f} \sum_{j=1}^{N_f} \Big( \Big( 3 - \Big( \tfrac{1}{N_d \, N_f^2} + \tfrac{1}{N_d^2 \, N_f} \Big) \\
& \qquad\qquad \sum_{p,q} k(\mathbf{h}_p^{(l-1)}, \mathbf{h}_q^{(l-1)}) \Big) \|\mu_{l-1}^{(d)} - \mu_{l-1}^{(f)}\| \\
& \qquad \left. + \|\mu^{(d)} - \mu^{(f)}\| + \left[ 2\sqrt{N} \Big( \Delta^l + \|\boldsymbol{\mu}_l^{(d)} - \boldsymbol{\mu}_l^{(f)}\| \Big) \right] \Big) \right]
\end{aligned}
\tag{30}
$$

We can rewrite it as:

$$
\Delta_{\mathrm{DP}} = \left| \frac{1}{N_d} \sum_{i \in \mathcal{S}_d} f(\mathbf{z}_i)_1 \; - \; \frac{1}{N_f} \sum_{j \in \mathcal{S}_f} f(\mathbf{z}_j)_1 \right|
$$

$$
\leq \left| f(\mathbf{z}_{\mu^{(d)}})_1 \; - \; f(\mathbf{z}_{\mu^{(f)}})_1 \right| \; + \; \frac{L}{2} \left( \frac{1}{N_d} \sum_{i=1}^{N_d} \| \mathbf{z}_i - \mathbf{z}_{\mu^{(d)}} \| + \frac{1}{N_f} \sum_{j=1}^{N_f} \| \mathbf{z}_j - \mathbf{z}_{\mu^{(f)}} \| \right)
$$

$$
\leq \left| f(\mathbf{z}_{\mu^{(d)}})_1 \; - \; f(\mathbf{z}_{\mu^{(f)}})_1 \right| \; + \; \frac{L}{2} \left( \frac{1}{N_d} \sum_{i=1}^{N_d} \| \mathbf{W}^{(l)} \| \mathbf{h}_{\overline{\mathbf{S}}D}^{(l)} + \frac{1}{N_f} \sum_{j=1}^{N_f} \| \mathbf{W}^{(l)} \| \mathbf{h}_{\overline{\mathbf{S}}D}^{(l)} \right) \tag{31}
$$

$$
= \left| f(\mathbf{z}_{\mu^{(d)}})_1 - f(\mathbf{z}_{\mu^{(f)}})_1 \right| \; + \; \frac{L}{2} \| \mathbf{W}^{(l)} \| \mathbf{h}_{\overline{\mathbf{S}}D}^{(l)} \left( \frac{N_d}{N_d} + \frac{N_f}{N_f} \right)
$$

$$
= \left| f(\mathbf{z}_{\mu^{(d)}})_1 - f(\mathbf{z}_{\mu^{(f)}})_1 \right| \; + \; L \, \| \mathbf{W}^{(l)} \| \, \mathbf{h}_{\overline{\mathbf{S}}D}^{(l)}
$$

Building on this analysis, we can derive an upper bound for the bias caused by sensitive-irrelevant node representations disparity.

$$
\Delta_{\mathrm{DP}} = \left| \frac{1}{|V_{S_d}|} \sum_{i \in \mathcal{S}_d} f(\mathbf{z}_i)_1 - \frac{1}{|V_{S_f}|} \sum_{j \in \mathcal{S}_f} f(\mathbf{z}_j)_1 \right|
$$

$$
\leq \left| f(\mathbf{z}_{\mu^{(d)}})_1 - f(\mathbf{z}_{\mu^{(f)}})_1 \right| \tag{32}
$$

$$
+ \; \frac{L}{2} \left( \frac{1}{|V_{S_d}|} \sum_{i=1}^{|V_{S_d}|} \| \mathbf{W}^{(l)} \| \mathbf{h}_{\overline{\mathbf{S}}D}^{(l)} + \frac{1}{|V_{S_f}|} \sum_{j=1}^{|V_{S_f}|} \| \mathbf{W}^{(l)} \| \mathbf{h}_{\overline{\mathbf{S}}D}^{(l)} \right)
$$

This completes the proof.

## C. Proof of Theorem 4.4

Assume there exist two distinct channels $\mathbf{h}_{c_i}$ and $\mathbf{h}_{c_j}$ ($i \neq j$) that both encode the sensitive attribute $S$. That is, $I(\mathbf{h}^{c_i}; S) > 0$ and $I(\mathbf{h}^{c_j}; S) > 0$.

Hence, we can write

$$
\mathbf{h}^{c_i} \; = \; f_i(S) + \varepsilon_i, \quad \mathbf{h}^{c_j} \; = \; f_j(S) + \varepsilon_j \tag{33}
$$

for some non-trivial functions $f_i(\cdot)$, $f_j(\cdot)$ and noise terms $\varepsilon_i, \varepsilon_j$.

Because both channels depend on $S$, we have

$$
I\big(\mathbf{h}^{c_i}; \mathbf{h}^{c_j}\big) \; = \; I\big(f_i(S) + \varepsilon_i; f_j(S) + \varepsilon_j\big) \; \geq \; I\big(f_i(S); f_j(S)\big) \; > \; 0 \tag{34}
$$

which contradicts the independence assumption $I(\mathbf{h}^{c_i}; \mathbf{h}^{c_j}) = 0$.

Therefore, under the requirement that all channels remain mutually independent, at most one channel can capture $S$.

## D. Experimental Settings

### Baselines

We compare FDGen with five state-of-the-art methods: performance-driven GRAPHARM (Kong et al., 2023) and fairness-aware models including FairAdj (Li et al., 2021), FG$^2$AN (Wang et al., 2023c), FairGen (Zheng et al., 2024), and FairWire (Kose & Shen, 2024b). A brief overview of these baselines is as follows:

- GRAPHARM: GRAPHARM is an autoregressive diffusion-based model for graph generation that directly operates in the discrete graph space by sequentially masking one node and its edges until the graph is empty. It employs a learned

node ordering strategy for more accurate likelihood approximation and achieves faster generation than existing graph diffusion models by limiting the number of diffusion steps to the number of nodes in the graph.

- FairAdj: FairAdj updates the normalized adjacency matrix to better satisfy dyadic fairness, which requires that link predictions be independent of the sensitive attributes from both vertices. This approach lessens the statistical gap between intra-group and inter-group link predictions while preserving as much predictive accuracy as possible.

- FG$^2$AN: FG$^2$AN is a fair graph generative model that addresses both node-level and structural fairness in creating synthetic graphs through adversarial training. It introduces tailored fairness metrics and a meta-strategy that reduces computational costs while handling multiple types of bias in the data.

- FairGen: FAIRGEN is a deep generative model that integrates label guidance and fairness objectives to produce synthetic graphs under limited labeled data. It uses a self-paced learning strategy and a novel context sampling approach to progressively learn the behaviors of protected and unprotected groups while preserving class memberships.

- FairWire: FairWire is a diffusion-based fair graph generation framework that uses a new fairness regularizer, to reduce structural bias in link prediction and synthetic graph creation. It captures the connections between synthetic sensitive attributes and the graph topology, allowing fair model training without revealing real sensitive data.

