# OpenReview forum: "FDGen: A Fairness-Aware Graph Generation Model"
_ICML.cc/2025/Conference — ICML 2025 poster_

### Official Review · Reviewer_Pk2M · 2025-03-08

**Overall Recommendation:** 2

**Summary:**

The authors propose FDGen a novel method for fair graph generation. The authors investigate the bias sources in graph generation, then consequently define regularization terms to promote fair graph generation by mitigating both the structural biases and node feature biases.

## update after rebuttal
My main concern regarding the very limited empirical gains of the proposed method remains valid. Figures 2 and 3 clearly highlight this issue. Therefore, I am maintaining my original score.

**Claims And Evidence:**

Yes, the claims made in the submission supported by clear and convincing evidence.

**Essential References Not Discussed:**

Essential related works are discussed and compared against.

**Experimental Designs Or Analyses:**

The experimental designs or analyses are sound.

**Methods And Evaluation Criteria:**

The proposed methods and evaluation criteria make sense for the problem at hand.

**Other Comments Or Suggestions:**

N/A

**Other Strengths And Weaknesses:**

I greatly appreciate the insights provided by the theoretical derivations.
My main concern is that the experimental results show that the proposed method performs roughly similar to comparison methods across all metrics and benchmark datasets. The fact that the insights of the theoretical derivations which inspired the design choices of FDGen did not translate into empirical gains against other graph generation methods which do not consider node feature biases is the main weakness of this work.

**Questions For Authors:**

N/A

**Relation To Broader Scientific Literature:**

The work is very relevant to the scientific literature. The theoretical derivations and investigations present valuable insights to the fair graph generation task.

**Theoretical Claims:**

Yes, theoretical claims and derivations seems sound to me.

---

> ### Author Rebuttal · Authors · 2025-03-30
>
> We thank Reviewer Pk2M for the time and thorough review. Below are our detailed responses:
>
> **The proposed method performs roughly similar to comparison methods across all metrics and benchmark datasets.**
> Our proposed method outperforms baselines with dissimilar performances. Specifically, in the graph generation tasks, the fair graph generation metrics, e.g., Fair-DD and Fair-Clus, consistently show our method performing better across all datasets. Our method achieves at least 10% improvement in fairness metrics across all four datasets compared to the baselines, with the most significant improvement of 25% on the Photo dataset. These improvements are achieved while maintaining comparable performance on quality metrics, e.g., DD and Clus. In addition, in downstream node classification tasks, our results also demonstrate substantial improvements. Compared to the worst fairness-aware baselines, our FDGen-GCN achieves a 25.61% improvement in fairness metrics. Even compared to the best baseline in fairness metrics, FDGen-GCN still achieves a 10.49% improvement in EO metrics. Moreover, compared to other fairness-aware baselines, these fairness gains come with 48.92% less accuracy loss, demonstrating our method's superior fairness-utility trade-off.

---

### Official Review · Reviewer_NfyT · 2025-03-08

**Overall Recommendation:** 4

**Summary:**

This paper proposes FDGen, a fairness-aware graph generation model that mitigates both structural and feature biases by introducing a fair regularizer and a diffusion-based framework to ensure fairness while preserving graph generation quality. Experiments on four real-world datasets show that FDGen outperforms SOTA methods in fairness and generation utility.

## update after rebuttal

**Claims And Evidence:**

This paper addresses an important problem in graph generation where both structural and feature biases can propagate through generated graphs and lead to unfair downstream decisions. Although fair graph generation has been studied previously, to my knowledge, this work is the first to address feature bias in the graph generation task, presenting a well-motivated approach with a solid theoretical foundation. It bridges the gap in existing fair graph generation research, which has primarily focused on structural bias.

**Essential References Not Discussed:**

Please refer to my comments above.

**Experimental Designs Or Analyses:**

I checked the experimental design and analyses for fairness and generation quality evaluations across four real-world datasets. The experiments are well-structured, and the results support the paper's claims. However, the clarity of the pictures and font size needs to be further improved.

**Methods And Evaluation Criteria:**

The proposed approach is technically sound. The fair regularizer effectively captures both feature and structural bias, while the diffusion-based generation framework maintains graph properties. The authors provide formal theoretical analysis with proofs, and their experiments on four diverse real-world datasets demonstrate consistent improvements across multiple fairness and quality metrics.

**Other Comments Or Suggestions:**

None

**Other Strengths And Weaknesses:**

Please refer to my comments above.

**Questions For Authors:**

Please refer to my comments above.

**Relation To Broader Scientific Literature:**

The paper advances the field of fair graph generation. The related work includes necessary and recent studies, covering fairness in graph learning and generative models. To my knowledge, this appears to be the first work addressing feature bias in graph generation. The paper extends prior research by tackling both structural and feature biases, offering a novel perspective on multi-source bias mitigation in graph generation.

**Theoretical Claims:**

I checked the theoretical claims and mathematical proofs, particularly regarding the fair regularizer and bias analysis in graph generation. The derivations are logically structured and well-motivated. While the proofs appear technically sound, additional clarification of underlying assumptions would strengthen their validity and applicability.

---

> ### Author Rebuttal · Authors · 2025-03-30
>
> We sincerely thank Reviewer NfyT for the detailed review and positive assessment of our work. We are particularly grateful for your recognition that FDGen is, to your knowledge, the first work addressing feature bias in graph generation, this was indeed a primary motivation for our research. We appreciate your thorough evaluation of our theoretical claims and experimental results, confirming the technical soundness of our approach in addressing both structural and feature biases. Your acknowledgment of our work's contribution to bridging an important gap in fair graph generation research is encouraging. We will improve the clarity of figures and font sizes as suggested. Thank you for recognizing the broader impact of our work in advancing fairness in graph generation models.

---

> > ### Comment · Reviewer_NfyT · 2025-04-05
> >
> > Thank you for the clarifications. I will keep my score.

---

> > > ### Author Response · Authors · 2025-04-05
> > >
> > > Dear Reviewer NfyT,
> > >
> > > Thank you for your positive decision and for taking the time to review our clarifications. We appreciate your feedback throughout this process. If you have any future questions about our paper, we're happy to address them.
> > >
> > > Best regards,
> > >
> > > Authors

---

### Official Review · Reviewer_LnUo · 2025-03-10

**Overall Recommendation:** 3

**Summary:**

The authors address fairness in graph generation problems, where fairness is meant as a trustful replication of the original graph that can then be used to train ML algorithm for automated decision making (e.g. credit scores). Their algorithm takes into account fairness both at the feature level and at the structural level. The authors designs a new fairness cost function that their algorithm minimizes.

**Claims And Evidence:**

yes

**Essential References Not Discussed:**

To the best of my knowledge there are not missing references

**Experimental Designs Or Analyses:**

The authors validate their algorithm over graphs which are totally irrelevant for this methodology. I would have expected some social network topology or something more related to social settings in the field of credit score and health. Instead they use datasets about amazon products or about paper citations.

**Methods And Evaluation Criteria:**

I feel I am missing something fundamental there.
Referring to section 4.3:
You need the original graph as input to the algorithm and then you produce a synthetic graph with is "fair" in the sense that you can faithfully replicate the feature and structural characteristics of the input. But if you need to have access to the original graph why don't you directly use it for training? And if not, how can you replace it?

**Other Comments Or Suggestions:**

Section 2.2, at the end you mention"However fairness remains unexplored in synthetic graph generation limiting these models'use in high stake scenarios": this is not exactly the motivation on why one should study fairness. I invite the authors to elaborate more on the importance of fairness when it comes to graph generation and bring more concrete examples on why fairness is important

In the notation:

Shouldn't the matrix A be in the graph definition as well? Namely $\mathcal{G}(\mathcal{V}, \mathcal{E}, X, A )$. Also, it is not clear to me what the difference between y and s is. Finally, "which includes \textbf{important} neighbor nodes of the central node", important is not a mathematical rigorous concept, rather define it in terms of "steps away" from the node?

Theorem 4.4 Reads more as an assumption than a theorem, indeed there is no proof for it

**Other Strengths And Weaknesses:**

Strength: relevant and novel work

Weaknesses: I am missing something fundamental, please refer to the "input graph" comment.
-The algorithm should be tested on more pertinent networks where the feature and structural biases are more clear

**Questions For Authors:**

-You need the original graph as input to the algorithm and then you produce a synthetic graph with is "fair" in the sense that you can faithfully replicate the feature and structural characteristics of the input. But if you need to have access to the original graph why don't you directly use it for training? And if not, how can you replace it?

**Relation To Broader Scientific Literature:**

I think in principle the problem is very important, namely how to avoid that graphs used for training ML algorithm are not biased against vulnerable communities which is very fundamental for credit score, health, etc.

**Theoretical Claims:**

No

---

> ### Author Rebuttal · Authors · 2025-03-30
>
> We appreciate Reviewer LnUo's thoughtful feedback and have provided responses below.
>
> **If you need to have access to the original graph why don't you directly use it for training? And if not, how can you replace it?**
> The original graph is not always suitable for training, so the generated graph is used instead. For example, in many real-world applications, organizations (like banks) possess valuable data but cannot directly share it due to privacy regulations. By generating synthetic graphs that preserve important information, we enable broader use of graph insights while maintaining privacy. Moreover, original graphs contain inherent biases. By using the original graph as input but applying our fairness constraints during generation, we create alternatives that reduce biases while maintaining utility, preventing discriminatory decision-making.
>
> **Selected datasets are fairness irrelevant.**
> The selected datasets are fairness relevant and widely used in fair graph research including fair graph generation. Specifically, citation networks (e.g., Cora and Citeseer) exhibit bias where papers from certain fields have higher visibility. For example, research on diseases predominant in white populations may receive more citations than equally important research on conditions affecting African American populations. Similarly, Amazon co-purchase networks (e.g., Photo and Computer) contain structural biases where purchase patterns can perpetuate stereotypes. For instance, if certain demographic groups historically purchase products associated with higher credit scores, individuals with different purchasing patterns may receive lower scores despite being financially responsible. In summary, due to the fairness relevance of these datasets, they have been widely adopted in fairness literature [1,2], including in our work.
>
> [1] Dong, et al. "Fairness in graph mining: A survey." TKDE 2023.
>
> [2] Zhang, et al. "Fairness amidst non‐IID graph data: A literature review." AI Magazine, 2025
>
> **Why is fairness important in generative models?** Fairness in generative models, particularly in the context of graph generation examined in this paper, is critically important; our work is motivated by the fact that synthetic graphs directly influence high-stakes decision-making across domains such as credit scoring. As shown in our toy example in Figure 1, unfair generation can amplify biases through structural bias (connections within the same sensitive groups) and feature bias (attribute disparities across groups). For instance, biased graphs can lead to biased loan approvals when male nodes are generated with higher income values or denser financial relationships. Without addressing both bias types, synthetic graphs replicate or even amplify bias, disadvantaging deprived groups in downstream applications.
>
> **Shouldn't the matrix A be in the graph definition as well?** The adjacency matrix A and edge set E are equivalent representations. Edge sets are more storage-efficient for sparse graphs. A 10,000 node graph would require 100 million matrix entries despite having only thousands of edges. Therefore, we define G={V,E,X}.
>
> **The difference between y and s.** s refers to sensitive attributes (e.g., gender), while y denotes node labels for downstream tasks (such as loan approval decisions).
>
> **Important is not a mathematical rigorous concept, rather define it in terms of "steps away" from the node?**
> Important in our work is mathematically defined through an importance score, which is based on both proximity and connection strength, rather than simply "steps away" from the node; Simply using steps away to build an ego graph may ignore truly important nodes or include too much noise information.
>
> **Theorem 4.4 is an assumption rather than a theorem and no proof.**
> Theorem 4.4 is a theorem rather than an assumption and here is the proof for reference.
>
> **Notations:** Let $h \in \mathbb{R}^{d}$ be node representation with channels $h = [h^{c_1},...,h^{c_N}]$. $I(\cdot)$ is mutual information; $I(X;Y)=0$ means independence.
>
> **Proposition.** For all $i \neq j$ with $I(\mathbf{h}^{c_i}; \mathbf{h}^{c_j}) = 0$, at most one channel can capture information about $S$.
>
> **Proof.** By contradiction. Assume at least two distinct channels $\mathbf{h}^{c_i}$ and $\mathbf{h}^{c_j}$ both capture information about $S$: $I(\mathbf{h}^{c_i}; S) > 0$ and $I(\mathbf{h}^{c_j}; S) > 0$. Then $\mathbf{h}^{c_i} = f_i(S) + \boldsymbol{\varepsilon}_i$, $\mathbf{h}^{c_j} = f_j(S) + \boldsymbol{\varepsilon}_j$ for nontrivial functions $f_i$, $f_j$ and noise terms $\boldsymbol{\varepsilon}_i$, $\boldsymbol{\varepsilon}_j$. Since both depend on $S$, $I(h^{c_i}; h^{c_j}) = I(f_i(S) + \varepsilon_i; f_j(S) + \varepsilon_j) \geq I(f_i(S); f_j(S)) > 0$, contradicting $I(\mathbf{h}^{c_i}; \mathbf{h}^{c_j}) = 0$. Thus, at most one channel can capture information about sensitive attribute $S$.

---

### Official Review · Reviewer_sM9W · 2025-03-13

**Overall Recommendation:** 3

**Summary:**

The authors propose a diffusion-based framework for fair graph generation that addresses both structural bias and feature bias within the generated graphs. Guided by theoretical analysis, which identifies how biases arise in the generation process, the framework applies a novel fairness regularizer to disentangle legitimate group differences from unfair biases, thereby preserving graph quality while ensuring fairness across different demographic groups. In an experimental study using four common graph datasets and five baseline methods, their approach improves fairness performance while maintaining generation quality.

**Claims And Evidence:**

Yes.

**Essential References Not Discussed:**

No

**Experimental Designs Or Analyses:**

Yes, it would be better to provide anonymous code.

**Methods And Evaluation Criteria:**

Yes.

**Other Comments Or Suggestions:**

1. This paper assumes that sensitive groups are clearly defined beforehand, but in practice, determining which attributes should be considered sensitive can be ambiguous and context-dependent.

2.  It would be better to provide anonymous code.

**Other Strengths And Weaknesses:**

1. The paper addresses an important yet often overlooked fairness challenge: feature bias in graph generation, providing well-motivated reasoning for its significance in real-world graph learning applications.

2. The theoretical analysis formally defines both feature and structural biases in node representations, offering analytical insights into how these biases propagate through graph generation processes.

**Questions For Authors:**

N/A

**Relation To Broader Scientific Literature:**

The paper builds upon prior work in fair graph learning and generative models by extending fairness research beyond structural bias to include feature bias, offering a theoretical analysis of bias propagation in graph generation and proposing FDGen as a novel mitigation approach.

**Theoretical Claims:**

The theorical proof is provided in the appendix.

---

> ### Author Rebuttal · Authors · 2025-03-30
>
> We sincerely appreciate Reviewer sM9W's thorough review and have provided detailed responses below.
>
> **Pre-defined sensitive attributes:** Our approach follows the standard convention in fairness research where sensitive attributes are predetermined based on legal frameworks and specific application contexts. In practice, sensitive attributes are typically defined by anti-discrimination laws (such as race, gender, age, and disability status under various civil rights regulations) or domain-specific ethical guidelines. For instance, in financial applications, factors like race and gender are legally protected categories, while in healthcare, additional attributes like genetic information may be considered sensitive. Our method is designed to work with these established definitions while remaining flexible enough to accommodate different sensitive attributes as required by specific applications. This assumption is consistent with most fairness literature in machine learning, as addressing the broader question of which attributes should be considered sensitive falls outside the scope of our technical contribution and belongs to legal and ethical domains.

---

### Decision · Program_Chairs · 2025-05-01

**Decision:**

Accept (poster)

**Comment:**

This paper introduces FDGen, a fairness-aware graph generation model that mitigates both structural and feature biases through the integration of a fairness regularizer and a diffusion-based generation framework. A key contribution lies in the theoretical analysis that formalises how biases emerge and propagated in the graph generation process and provides analytical tools to disentangle legitimate group differences from unfair biases. This enables the model to preserve graph quality while improving fairness across demographic groups.

The theoretical contribution is well recognised and appreciated by all reviewers, particularly the formal definition and propagation analysis of structural and feature biases in node representations. These insights add significant value to the growing body of work at the intersection of fairness and generative models.

However, reviewer opinions diverged on the empirical results. Additionally, reviewers noted that parts of the paper—particularly the core methodology and technical formulation—could benefit from clearer exposition to ensure accessibility for a broader audience, especially those less familiar with fairness-aware graph learning.